# Inward motion of diamond nanoparticles inside an iron crystal

Yuecun Wang [1,5], Xudong Wang[2,5], Jun Ding[2,5], Beiming Liang[1], Lingling Zuo[1], Shaochuan Zheng[1], Longchao Huang [1], Wei Xu [1], Chuanwei Fan[1], Zhanqiang Duan[3], Chunde Jia[3], Rui Zheng[1], Zhang Liu[1], Wei Zhang [2], Ju Li [4], En Ma [2] ✉ & Zhiwei Shan [1] ✉

In the absence of externally applied mechanical loading, it would seem counterintuitive that a solid particle sitting on the surface of another solid could not only sink into the latter, but also continue its rigid-body motion towards the interior, reaching a depth as distant as thousands of times the particle diameter. Here, we demonstrate such a case using in situ microscopic as well as bulk experiments, in which diamond nanoparticles ~100 nm in size move into iron up to millimeter depth, at a temperature about half of the melting point of iron. Each diamond nanoparticle is nudged as a whole, in a displacive motion towards the iron interior, due to a local stress induced by the accumulation of iron atoms diffusing around the particle via a short and easy interfacial channel. Our discovery underscores an unusual mass transport mode in solids, in addition to the familiar diffusion of individual atoms.

Carbon is the most important alloying element commonly added into iron for making steels[1,2], and also widely exploited for surface treatment: carburizing, known as case hardening, has been utilized for centuries[1]. In carburizing, upon heating individual carbon atoms from a surface source diffuse into the iron lattice through atomic diffusion[2], dissolving as interstitials[3] and often precipitating out as cementite[4] or graphite[5]. However, in all these familiar cases, carbon never appears in the form of *diamond*, which is the most desirable allotrope of carbon and an attractive reinforcement of metallic materials due to its superior mechanical strength[6–8], thermal and chemical stability[9], low friction[10] and thermal expansion coefficients[11], as well as high thermal conductivity[12]. Compared with the formation of much easier and routinely observed cementite or graphite, nucleating and growing diamond inside a carburizing iron crystal is impracticable[13]. One then wonders if it is ever possible for a particulate diamond, such as rigid diamond nanoparticles, to migrate as a whole into the interior of solid iron or steels from the outside, above and beyond conventional solid-state diffusion, which is well known to be mediated by interstitial or substitutional diffusion of individual atoms.

In the following, we demonstrate that the above rigid-body motion can be realized, using a series of experiments. We show that the diamond nanoparticles (DNPs) can indeed become embedded into iron or steels and move inside, at temperatures of $0.4 \, T_m$ – $0.6 \, T_m$ ($T_m = 1811$ K is the melting point of Fe). The directional motion distance of DNPs reaches as far as ~millimeter in depth. Our observation is first made upon monitoring carburizing in action through in situ scanning electron microscope (SEM) and transmission electron microscope (TEM) experiments. We then examine how the process played out in an industrial setting—furnace carburizing of a bulk steel. In the ensuing section, we propose a mechanism that can sustain the DNP motion inside the iron crystal, in particular the available thermodynamic driving force and kinetics, without the full graphitization or dissolution of the DNPs.

[1]Center for Advancing Materials Performance from the Nanoscale (CAMP-Nano) & Hysitron Applied Research Center in China (HARCC), State Key Laboratory for Mechanical Behavior of Materials, Xi'an Jiaotong University, Xi'an 710049, China. [2]Center for Alloy Innovation and Design (CAID), State Key Laboratory for Mechanical Behavior of Materials, Xi'an Jiaotong University, Xi'an 710049, China. [3]Department of Materials Science and Engineering, Shenyang Ligong University, Shenyang 1100159, China. [4]Department of Nuclear Science and Engineering, and Department of Materials Science and Engineering, Massachusetts Institute of Technology, Cambridge, MA 02139, USA. [5]These authors contributed equally: Yuecun Wang, Xudong Wang, Jun Ding. ✉e-mail: maen@xjtu.edu.cn; zwshan@xjtu.edu.cn

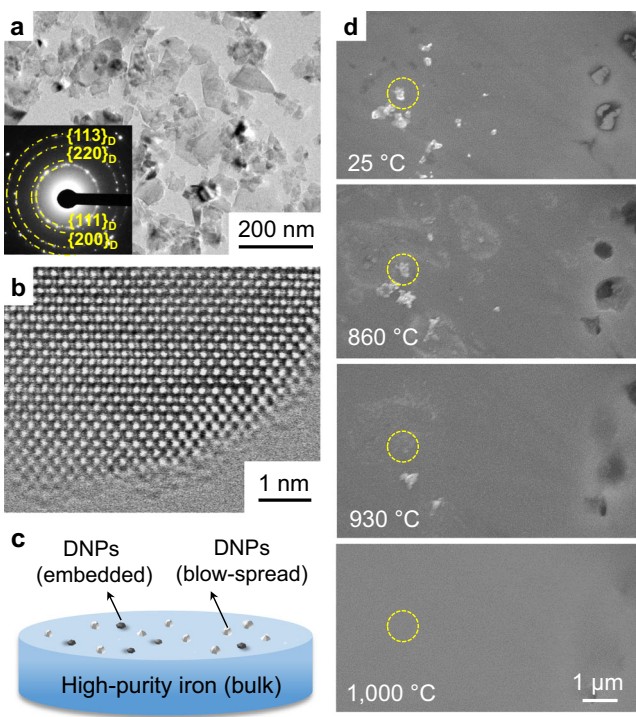

**Fig. 1 | In situ SEM observation of the sink-in of diamond nanoparticles (DNPs) into iron. a** Typical TEM image together with the selected area electron diffraction pattern (inset) of the nanoparticles in diamond powders. **b** The atomic TEM image of a DNP. **c** Schematic lay-out of the in situ SEM heating experiment. DNP aggregates (white spots) were blow-spread on the surface of bulk iron. The DNPs (gray spots) partly embedded into the surface resulted from the fine polishing of the iron piece using diamond powders. **d** SEM top view of scattered DNPs on surface or those partially embedded into the surface getting gradually buried into iron with increased temperatures, eventually leaving a flattened iron surface. The dash yellow circle highlights the position of one of DNP aggregates sitting on surface.

## Results

### In situ SEM and TEM observations

The diamond powders we used were aggregates, each consisting of agglomerated DNPs that are typically below ~100 nm in size (Fig. 1a). The DNP aggregates have been confirmed to be mainly of the diamond cubic crystal structure with $sp^3$-hybridized bonding, via atomic-scale TEM imaging (Fig. 1b), electron energy loss spectroscopy (EELS) and Raman spectroscopy (Supplementary Fig. 1). Details of the samples and the characterizations are presented in Supplementary Information.

To monitor the process of DNPs moving toward the interior of iron, two assemblies of DNPs/iron crystal have been designed: some DNPs (gray spots) are embedded (during grinding) into the bulk iron surface and some DNPs (white spots) are blow-spread on the iron surface (see the lay-out in Fig. 1c). Real-time SEM observations gave the first indication of the DNPs getting gradually buried into the bulk iron underneath as temperature increases. This is recorded in a series of SEM micrographs in Fig. 1d. DNPs embedded partially into the surface (dark spots) submerge into the iron surface, and in the meantime, those sitting on surface (a DNP aggregate is highlighted by the yellow circle) also sink into the iron substrate bit by bit, especially dramatic at temperatures above 800 °C. All the DNPs have completely sunk into the iron substrate at 1000 °C, leaving behind only a clean and smooth surface. Raman spectroscopy scans, performed on the in situ heated and smoothened iron surface (after ultrasonic cleaning and polishing), affirm the presence of diamond phase inside iron (Supplementary Fig. 2) as the signals (peak at 1326 ~ 1332 cm⁻¹) provide clear identification of nanodiamond[14,15]. Note that there are also minor signals from nanocrystalline or disordered graphite at 1350 cm⁻¹ (D band)[16] and 1600 cm⁻¹

(G band)[17]. This is an indication that a small portion of the DNP has been graphitized (to be discussed further later).

In order to unravel the sink-in process of DNPs more clearly, in situ experiments were carried out inside a TEM (see experimental details in "Methods" section). Figure 2a presents a series of TEM snapshots from the in situ Movie, taken in a side view showing a DNP aggregate sitting on the surface of an iron foil. At room temperature (RT), there is a thin native oxide scale (~8 nm, marked by a green line) on the iron surface. When heated to ~400 °C, the oxide scale in contact with DNPs gets reduced by carbon, exposing fresh iron to DNPs. At ~500 °C, the oxide scale is gone. Meanwhile, the flat outer facet of iron becomes rugged, and the projected view of iron gets darker, suggesting gradually increased thickness due to the arriving Fe via rapid surface diffusion, toward the location in contact with DNPs. As the temperature increases, the DNP aggregate appears to be consolidating itself and attracted to iron, attaching intimately onto the latter. In the meantime, a flux of Fe flows toward the top surface, via fast surface diffusion to wrap around the DNPs. The continued up-flow of Fe eventually engulfs DNPs altogether. The entire dynamic process of this initial sinking-in is shown in Supplementary Movie 1 and Supplementary Movie 2. Post-mortem characterizations after cooling down to RT confirm the existence of DNPs inside iron (Supplementary Fig. 3). EELS spectrum (Fig. 2c) from the square box in Fig. 2b (another DNP aggregate that was not engulfed) exhibits strong signals of Fe. This validates that fast surface diffusion has apparently flown a sufficient quantity of Fe atoms to wrap around DNPs. High-resolution scanning TEM (STEM) image shows that the surface of a DNP has been covered by a thin layer of graphite before it is engulfed into iron (Fig. 2d). The surface graphitization of DNPs is catalyzed by the arriving Fe flux[18].

### DNPs into the bulk steel in furnace

To observe if and how the above process plays out in the conventional heat treatment, we now move on to the experiment with bulk low-carbon steels inside a tube furnace. After heating at ~980 °C in a sealed container or the argon atmosphere for several hours, followed by furnace cooling down to RT, the specimen was thoroughly cleaned, ground and deeply etched to investigate the sizes and distribution of DNPs in it (Fig. 3b, left). For a typical example (treated at ~980 °C for 5 h), the SEM image (Fig. 3b, right) shows a high density of dispersed nanoparticles at a depth of ~0.05 mm into the sample. They have been validated to contain diamonds via the Raman spectrum analysis (Fig. 3c). These nanoparticles display variable sizes, because DNPs are of different original sizes to begin with. When the heat treatment time increases to 24 h, Raman signals from nanodiamond can be detected at a depth as far as ~2 mm (Fig. 3c). Our results indicate that DNPs can enter deep into the steel, and their average size decreases (due to dissolution of the surface graphitized layer) with increasing distance in the depth range we investigated (Supplementary Fig. 4). To further demonstrate DNPs in the interior, we dissolved the iron matrix away in hydrochloric acid. TEM characterizations confirmed that the remnants indeed contain many DNPs with a small number of graphite nanoribbons (Supplementary Fig. 5). The interface between a DNP and the iron matrix is found to be semi-coherent (Fig. 3e) and has the tendency to form the $\{110\}_{Fe}$-$\{111\}_D$ crystallographic relationship (also see Supplementary Fig. 6). Chemical bonding analysis via density functional theory (DFT) calculations (Supplementary Fig. 7) of such interfaces indicate well-bonded Fe-C at the interface.

## Discussion

The initial sink-in stage and the ensuing inward motion of a DNP inside the iron crystal are shown schematically in Fig. 4a. When heated to a critical temperature, the iron oxide scale begins to decompose. Freshly exposed Fe atoms flow from underneath the DNP and wrap around it via fast surface diffusion (see Supplementary Discussions Section I, Supplementary Fig. 8 and Supplementary Movie 3). This action can be

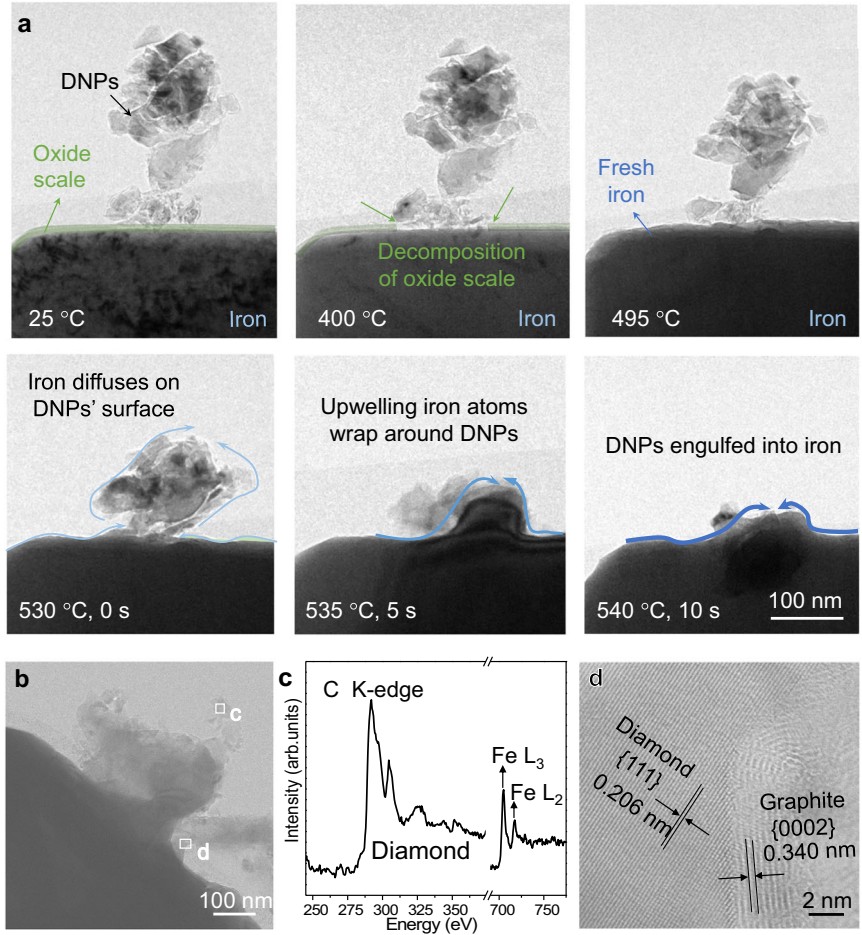

**Fig. 2 | In situ TEM observation of diamond nanoparticles (DNPs) being engulfed into iron. a** A typical example showing the engulfed process of a DNP aggregate on the lateral surface of pure iron. The oxide scale on surface of iron is highlighted in green. The blue curves with arrows represent the Fe flux. **b** STEM image of another DNP aggregate before it was fully engulfed into iron. **c** EELS spectra acquired from the square box "c" in (**b**). The C K-edge corresponds to the second K-shell ionization loss peak of crystalline diamond at 299 eV, and the characteristic Fe L$_2$ and Fe L$_3$ edges prove the existence of Fe. **d** High-resolution STEM image taken from the "d" box in (**b**). Note the graphite layer on the outer surface.

construed as Fe striving to cover up or wet carbon to lower the surface energy of the DNP. In this stage (Fig. 4a, I), a capillary force arises from the interaction between the Fe flux and the DNP surface, directed along the DNP-Fe interface, driving the particle toward the inside. The resultant stress at the bottom interface of Fe-DNP can reach gigapascal level, according to the burrowing model proposed by Zimmermann et al.[19,20]. Meanwhile, the surface atomic layers of the DNP get graphitized gradually, in the presence of Fe, which is known to catalyze graphitization[18]. After the DNP is buried inside solid iron (Fig. 4a, II), however, all its surfaces or interfaces are completely surrounded by the iron matrix and the net capillary force to drive the directional motion of the particle diminishes. Therefore, there must be other driving forces to propel the continued downward motion of the DNP.

We first show that there is a thermodynamic driving force available to sustain the upward mass transport of Fe. It has been reported that small inclusions can move inside the solids driven by the gradient of chemical potential[21]. The Gibbs free energy versus composition diagram for Fe-C is displayed in Supplementary Fig. 9. The free energy curve predicts an equilibrium solubility at ~7 at.% C (common tangent touch point, see Supplementary Fig. 9). Normally the intermixing action tending toward thermodynamic equilibrium is carried by the interstitial diffusion of C atoms into Fe[22], as the outbound substitutional Fe atoms diffusion through crystal lattice is far slower. A carbon concentration gradient is expected from the surface toward the interior of bulk Fe. In our case of submerged DNPs (Fig. 4a II, III), a

carbon concentration gradient is also built up: the closer to the top surface, the higher the carbon concentration, as evidenced by the depth profiles and three-dimensional-compositional images of C and Fe from the time-of-flight secondary-ion mass spectroscopy analysis (ToF-SIMS, Fig. 4b). But here the source of C is the graphitized surface layer of each and every DNP gradually dissolving to satisfy the solubility in the surrounding Fe. The carbon concentration profile could be maintained for a long time via the continual supply from these DNPs (the number of DNPs entering Fe is sufficiently large). In the meantime, a Fe concentration gradient is also established, but in the opposite direction. Note that different from the conventional carburizing, in which the carbon source only exists outside the steel and the carbon concentration gradient becomes shallower and shallower along the millimeter depth direction, here each engulfed DNP with a gradually graphitized and meanwhile dissolving interfacial layer acts as a movable source of carbon by itself. As a result, the large carbon concentration gradient extends forward along with the moving DNPs (inhomogeneous concentration field). Take the leading DNP as an example, as it moves forward deeper and deeper, it keeps encountering iron interior with less carbon solute content (i.e., the DNP always sees a higher Fe concentration at the location in front of its moving trajectory, Fig. 4a III, IV). The dissolved carbon atoms at the location underneath the DNP diffuse into the "virgin" iron more rapidly than into the already-carbon-enriched locations above the DNP. That is, in its surroundings the DNP is always accompanied by a local

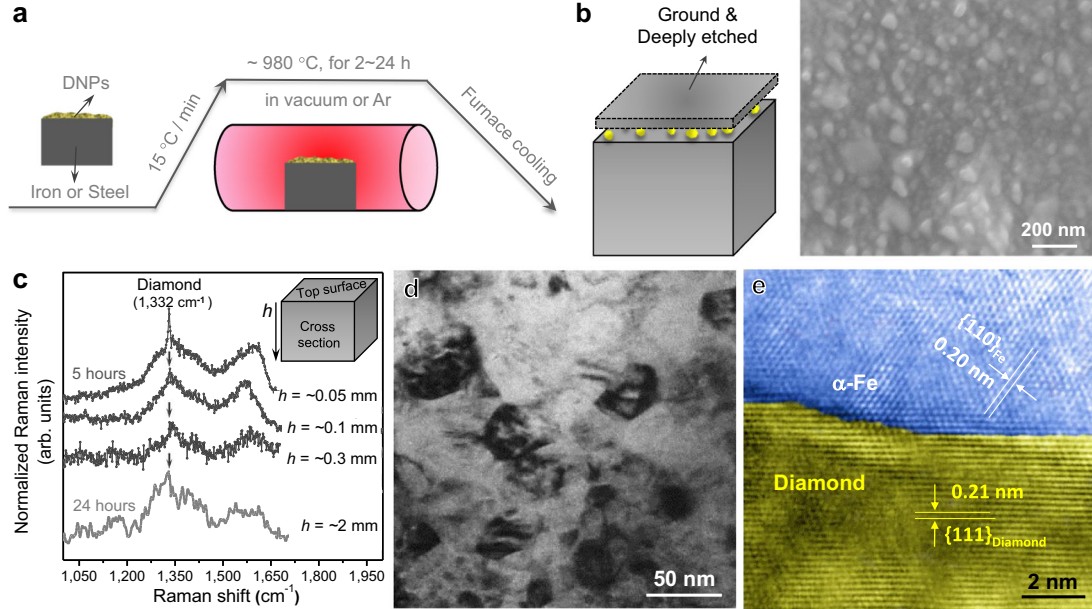

**Fig. 3 | Diamond nanoparticles (DNPs) move into bulk steel up to millimeter depth. a** Schematic showing the experimental setup in furnace. **b** Schematic illustration (left) and the SEM image (top view, right) of DNPs exposed, after the top surface was ground and etched away. **c** Raman spectra acquired at different depths (*h*, see the inset) of -0.05 mm, -0.1 mm and -0.3 mm from the cross section of a sample treated in furnace for 5 h and that acquired at the -2 mm depth of a sample treated for 24 h. Characteristic peaks at 1332 cm⁻¹ prove the existence of diamond inside iron at different depths, and meanwhile the G band of graphite at 1560 - 1580 cm⁻¹ can be found. **d** A TEM image of the dispersed DNPs inside iron. **e** High-resolution TEM image of the interface between a DNP and the iron matrix. The upper part in blue shows the lattice of α-Fe (the interspace of {110} atoms is about 0.20 nm), and the lower part in yellow is the lattice of diamond (the interplanar distance of {111} is about 0.21 nm).

concentration gradient, see the C profile in a local (smaller volume) region, likely around some DNPs, as displayed in the inset in Fig. 4b.

In the presence of the Fe concentration gradient, as schematically illustrated in Fig. 4a, a Fe flux flows in the direction from the location underneath the DNP (higher Fe concentration and higher chemical potential) toward its interfacial boundary with Fe above the DNP (higher carbon concentration and lower chemical potential). As some space is "opened up" by the leaving Fe atoms that constitute the upward flow, the side underneath the DNP becomes the "looser spot". Meanwhile, a local stress arises on the side above the DNP, when the arriving excess Fe atoms are trying to squeeze into the limited space at the upper DNP/Fe interface. This local stress pushes the DNP to take a downward rigid-body translation into the bulk Fe. Such diffusion-induced stress has been established in solid-state interdiffusion[23], electromigration[24,25] as well as thermomigration[26] processes. In the temperature gradient case as an example[26], metal (such as Al, Ag, Cu, etc.) atoms diffuse into a narrow zone sandwiched by two solid bodies. In this zone the incoming metal flux has to take additional space, causing a traffic jam inside it. This results in a stress that drives the translational movement of the metallic crystals, and the stress magnitude is estimated up to be -10² MPa[26], which is of the same order of magnitude as our estimate of the local stress at the upper DNP/Fe interface (see Supplementary Discussion III for details). Therefore, regardless of the type of driving gradient (chemical composition gradient, temperature gradient or electrical potential gradient) as long as there is a net diffusional flow, in the vicinity of the location/side of net mass accumulation, a mechanical stress is generated. This local stress nudges the DNP as a whole to move toward the side where the Fe flux originates from. Note that the downward motion of the DNP may leave behind an "easy channel" that could further promote the upward fast diffusion of Fe flux toward the sample surface. Fe atoms leaving from the upper interface of Fe/DNP help to sustain the local concentration gradient around the DNP and hence continuous motion of the particle.

Kinetically, the upward flow of Fe is faster than the downward carbon flux for the Fe-C intermixing. This Fe relocation is sufficiently fast because the Fe atoms only need to travel a very short distance around the tiny DNP. Fe has been known to readily catalyze graphitization of diamond[18,27,28], and the migration energy of Fe on graphite is only 0.0068 eV[29]. In the absence of open graphite surface, the graphitized atomic layers on the DNP surface can offer an easy migration channel for Fe diffusion. The latter is shown in Fig. 4c, where our Nudged Elastic Band (NEB) DFT calculations (see Supplementary Fig. 7 and "Methods" section for details) suggest that the energy barrier for Fe migration is 0.69 eV for the first gap with the outmost (0002) atomic plane bonded with Fe, and 0.28 eV for the next gap in between the neighboring (0002) layers. Such activation barriers are much lower than the -1.5 eV[30] of the interstitial diffusion of C in Fe lattice, or the -3 eV for self-diffusion of Fe atoms[30]. Therefore, the easiest kinetic path for Fe diffusion is the gap in between the neighboring (0002) layers with an interplanar spacing of -0.34 nm, which offers extra space for the migrating Fe atoms (0.25 nm in diameter) that can enter graphite layers from their flaws or defect sites. Estimates (see "Methods" section) indicate that even for the lower-bound scenario (assuming the higher 0.69 eV energy barrier) the Fe flow would be adequate to account for the experimentally observed DNP motion velocity (*v*). Also note that Fe has little solubility in graphite (see comment tangent in Supplementary Fig. 9), so that its accumulation inside the thin graphite layer would be very limited. Instead, to equilibrate the chemical potential difference the Fe atoms travel to the top DNP interface to mix with the gradually dissolving outmost graphite atomic plane, expanding the matrix above and shifting down that interface. As such, the parallel graphite atomic layers merely serve as a diffusion channel to facilitate Fe transport.

Monte Carlo (MC) simulations lend support to the mechanism proposed above. Figure 4d plots the MC-derived location of a DNP versus time under various chemical potential gradient (∇μ) levels. It

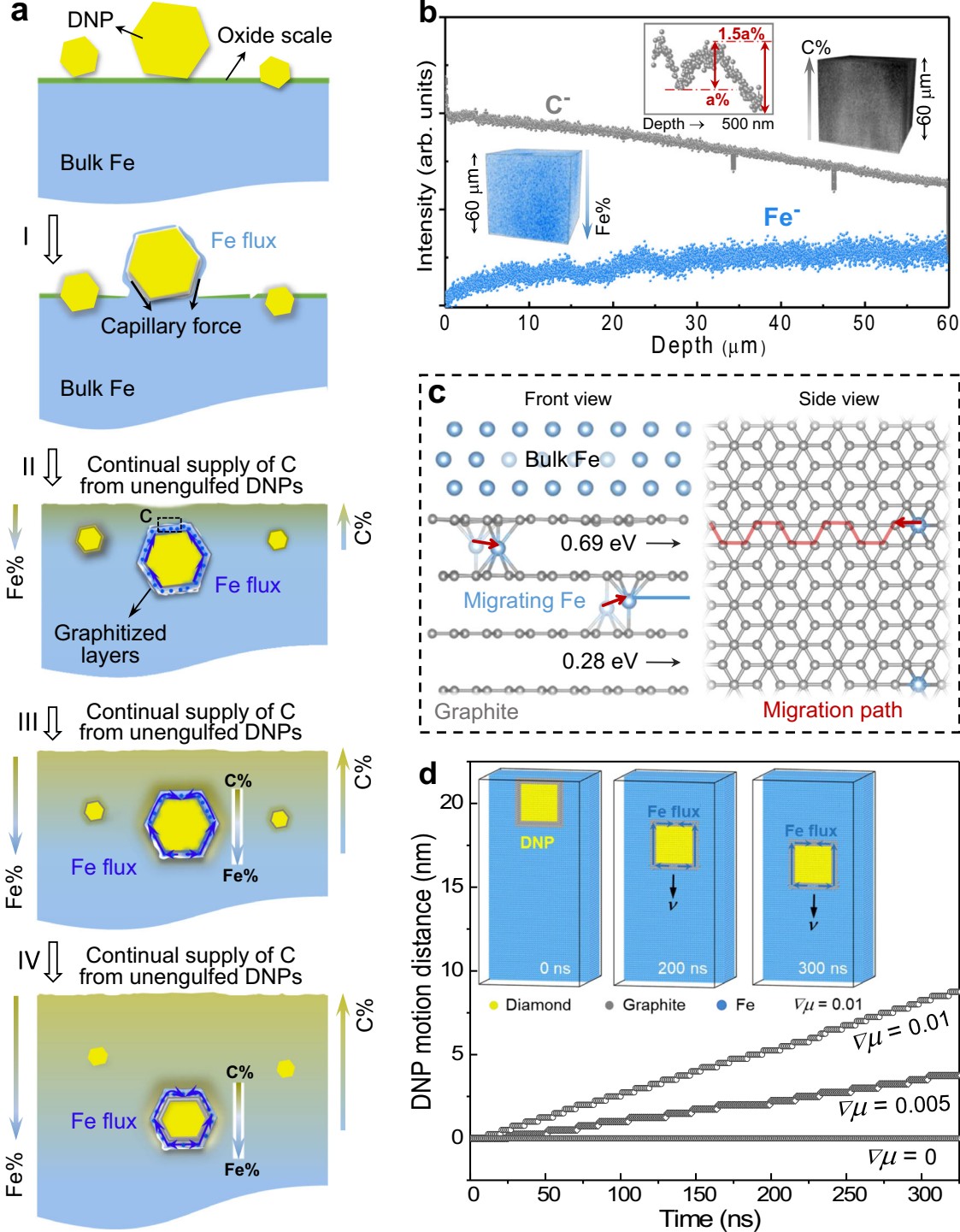

**Fig. 4 | Mechanism for the inward motion of diamond nanoparticle (DNP) into the iron crystal. a** Schematic depicting the steps involved in the process. DNPs, Fe and graphite are in yellow, bule and gray, respectively. The oxide scale on Fe surface is marked in green. The lines with arrows on the left and right sides show the opposite directions of the Fe concentration (Fe%) and C concentration (C%) increase, respectively. **b** ToF-SIMS analysis of a sample heated in furnace for 1 h (all dispersed DNPs have entered the iron matrix) and then quenched in water to retain the carbon distribution at high temperatures as much as possible. The ToF-SIMS spectra showing the depth profiles of secondary ions of $C^-$ (gray) and $Fe^-$ (blue) in the sputtered volume from the top surface to a ~60 μm depth over an area of ~50 × 50 μm². The corresponding 3D images of the depth profiles visualizing the opposite concentration gradients of C and Fe. The inset ToF-SIMS spectrum

displays the depth profile of $C^-$ in a much smaller volume (the analyzed area is ~5 × 5 μm² and the depth is ~500 nm). **c** The migration path and energy barrier predicted by NEB analysis, for the diffusional hopping of Fe atom along the channel in-between graphite (0002) planes. The energy barrier for Fe migration is 0.69 eV in the first gap with one (0002) atomic plane bonded with Fe, and 0.28 eV for the next gap in between the neighboring (0002) layers, see "Methods". **d** The motion distance of DNP as a function of time under different chemical potential gradients ($\nabla\mu$, in eV per interatomic distance), obtained via Monte Carlo (MC) simulation (see "Methods") at 1200 K for the 0.28 eV case. The inset shows the MC-simulated DNP motion under $\nabla\mu = 0.01$, where Fe flux plating into the upper interface propels the downward translation of the DNP at velocity $v$ (see Supplementary Movie 4).

was found the larger is the $\nabla\mu$, the farther a DNP can move in a given time. Figure 4d displays atomic configurations showing a DNP motion under $\nabla\mu = 0.01$ (eV per interatomic distance), where Fe flux plating into the upper interface propels the downward translation of the DNP at $v$ (see Supplementary Movie 4). Note that the velocity in Fig. 4d far exceeds the experimental $v$ and can only serve as a prediction of the trend, because unrealistic parameters were used as input to allow observable DNP motion on the time scale of the computation. An extrapolation/estimate to enable comparison with the measured $v$ in experiment and the calculated $v$ analytically is discussed in Supplementary Discussions Section II (Supplementary Fig. 10) and Section III.

The downward movement speed is projected to decrease as the DNP gets sufficiently deep into iron. This happens when the DNP size decreases to below a critical diameter due to the gradual dissolution of the graphite layer on its surface. While the graphite layer is partly replenished, via dynamic graphitization of the outmost atomic layers of the DNP in the presence of the catalyzing iron, this graphitization would eventually become difficult. This is because the penalty increases as the DNP size decreases: the graphitization requires a volume expansion against the iron matrix, causing a high pressure that would make the transformation from diamond to graphite energetically unfavorable (see Supplementary Fig. 11 and Supplementary Discussions Section IV for more details). In the absence of the graphite layer, the interface between Fe and DNP becomes semi-coherent and atomically sharp (Fig. 3e), the easy channel for Fe migration vanishes, and the moving velocity of DNPs diminishes.

That a rigid-body mechanical motion can be induced by a chemical driving force is known from the Kirkendall effect[23]. Our case here, however, has significant differences. First, instead of chemically inert wires as fiducial markers, here our DNP "fiducial markers" are actively dissolving, thus creating the concentration gradient as the root cause of the directional iron diffusion along the interface. Second, instead of the lattice vacancy exchange/accumulation mechanism as in the Kirkendall experiment, here it is the interfacial diffusion of Fe that generates local stresses to induce the rigid translation of particles. The Fe concentration gradient in our case is set up by carbon dissolution, driving the interfacial-diffusional flow of Fe, and subsequently the rigid translational motion of the DNP "markers".

The mechanism proposed above is more plausible than other alternatives we can find out at present, and two of them are discussed in the following. First, it has been reported that migrating grain boundaries can drag with them some solid or liquid small oxide particles inside the metal matrix, causing the accumulation of particles on the grain boundary[31,32]. In our case, no obvious aggregation of DNPs on the grain boundary or DNPs-denuded region near the grain boundary has been found (see Supplementary Fig. 12). Instead, the DNPs are found spread-out inside the Fe matrix. Second, one may argue that DNPs inside the iron interior come from the dissolution of the original DNPs, followed by the diffusion and then segregation of supersaturated carbon atoms to re-precipitate diamond particles upon cooling. However, this diffusional process relying on individual atoms would be the same as that in the conventional carburizing treatment, which never produced diamond inside the steels. What's more, even with supersaturation and segregation, the precipitation of diamond is far more difficult than the nucleation of the routinely observed graphite and cementite: the minimum temperature/pressure required for diamond formation in the Fe-C system would be about 1200 °C/5 GPa[33].

In terms of fundamental materials science, our observation points to a mass transport mode for the motion of particulate entities inside a solid while maintaining their crystal structure, aside from the individual jumps that mediate normal atomic diffusion. The finding connects solid-phase dissolution, coarsening or precipitation, which only consider diffusion of individual atoms, with mechanical translation.

The concentration-gradient-driven mass relocation in our case differs from creep where the chemical potential gradient for the diffusional mass transport is generated by an applied macroscopic mechanical stress. In terms of practical applications, the graded metal surface on millimeter depth scale with programmable and well-dispersed nanoparticles distribution holds the promise for opening an avenue for previously unattainable properties. As a specific example, the introduction of DNPs into metallic matrix yields superior mechanical properties. For instance, the quenched Fe-DNPs sample has remarkably improved hardness over some typical case-hardening alloy steels[34,35] as well as significantly decreased coefficient of friction (Supplementary Fig. 13). In terms of the broader applicability of our transport mode, metallic or ceramic nanoparticles other than DNPs can be deposition-coated with a diamond shell with a tunable thickness[36,37], such that they can also be introduced into Fe or steels. The translational transport of nanoscale particulate matter opens an easier way for creating gradient composites with enhanced near-surface properties.

## Methods

### Diamond powders
DNP powders were prepared using high-energy ball milling of high-pressure high-temperature (HPHT) diamond microcrystals (Henan Yuxing Micron Diamond Co. Ltd., China, ~US \$280/kg). Raman spectroscopy characterization (Supplementary Fig. 1a) shows the well-documented diamond peak at ~1330 cm$^{-1}$, some peaks from nano-crystalline or disordered sp$^3$-hybridized carbon at 1160 cm$^{-1}$ and 1406 cm$^{-1}$[16], as well as the sp$^2$-hybridized G band of amorphous carbon at 1550 cm$^{-1}$[14], which may result from the high-energy ball milling. The low-loss and core-loss EELS of DNPs (Supplementary Fig. 1b) demonstrate the characteristic plasmon peak with the energy loss value of ~34 eV and K-edge of diamond[38]. Noted that a very thin amorphous carbon, mainly existing on the surface of DNPs (Supplementary Fig. 1c), helps to (1) accelerate the reduction of oxide scale on iron[39], facilitating the surface diffusion of Fe atoms and the sinking-in of DNP into iron; (2) decrease the graphitization temperature of DNPs in contact with Fe (from 1000 °C to 500-600 °C)[40,41]; and (3) establish a carbon concentration gradient upon its dissolution into the iron matrix.

### Raman spectroscopic experiments
Raman spectroscopic experiments were carried out using Horiba Jobin-Yvon LabRam HR800 with the 1.5 cm$^{-1}$ spectral resolution and 1 μm spatial resolution. The incident Ar$^+$ laser wavelength is 532 nm, and the output laser power is 100 mW. The iron/steel samples with DNPs inside were deeply etched using 4 vol% Nital (solution of nitric acid and ethanol) or focused ion beam milling to expose the particles beneath the surface of iron. To locate the DNPs, a zone was carved out using FIB milling. This marked zone was large enough and can be easily found under optical microscope. Then we blew diamond powders over the sample. Raman spectroscopic scans (Supplementary Fig. 2) were performed on two regions (FIB-marked) of the specimen, which was heated to 1000 °C inside SEM, cooled down to RT and followed by ultrasonic as well as ion beam cleaning. The size of each scanned region was 12 μm × 12 μm, generating two 6 × 6 datasets. Since Raman scattering intensity is proportional to the analyte concentration[15], the normalized intensity of diamond characteristic peak (~1330 cm$^{-1}$) acquired from each spectrum can reflect the concentration of DNPs in the local region. We mapped out the Raman intensity distributions, which turned out to be roughly consistent with the locations where the original DNPs resided before heating, suggesting the vertical entrance of DNPs into iron. However, nanodiamond signals can also be detected from some areas without DNPs before, which indicates that some DNPs disaggregate into scattered smaller clusters or individual particles during their downward motion inside iron.

### In situ SEM heating experiments

DNPs were blow-spread over the surface of mirror polished high-purity iron (99.995%, Alfa Aesar, Thermo Fisher Scientific Inc.). Present at the same time are some DNPs (average particle size of 500 nm), partly embedded into the surface, which were the residues left behind from fine polishing of the iron piece using DNPs. The powder/substrate assembly above was placed onto an 8 mm alumina disk of the in situ SEM heating instrument[42], which is capable of reaching temperatures as high as 1150 °C. The hot plate is large enough to ensure a uniform temperature distribution on the sample surface. Thermal electrons are blocked by a metal shield for observing the dynamic evolution of the surface clearly under electron beam scanning. The surfaces were then monitored during heating from room temperature to 1000 °C (over a time period of less than 2 h) inside a SEM (S8000, TESCAN, Czech Republic).

### TEM and STEM characterizations

TEM/STEM observations, electron diffraction and EELS were conducted using a high-resolution TEM (JEM-2100F, JEOL®, Japan, operating at 200 kV) as well as a Cs-corrected S/TEM (HF5000, Hitachi® High-Tech, Japan, operating at 200 kV).

### In situ TEM heating experiments

In situ heating tests were performed inside the environmental TEM (E-TEM, H9500, Hitachi® High-Tech, Japan, performed at 300 kV) using a home-made micro-electromechanical systems (MEMS) heating device (Supplementary Fig. 3a), which has accurate temperature control and minimized thermal drift[43], affording the high spatial resolution needed to observe what actually is going on at the atomic level. A specimen was lifted out from the high-purity iron and welded onto one of the mounting bars of the heater via platinum deposition inside the FIB chamber (Supplementary Fig. 3b). DNPs were scattered on the lamella surface and then heated from room temperature to high temperatures gradually.

### Bulk steel samples heated in furnace

The raw material used for furnace carburizing was low-carbon (0.15 at.% C) steel, which was cut into specimens with dimensions of 10 mm × 10 mm × (5 - 20) mm. The top surfaces were ground, mirror polished and cleaned. The well-dispersed nanodiamonds solution (after ultrasonic dispersion with an ultrasonic bath for at least 2 h) was smeared evenly overlying the specimen, onto which an iron or steel thin plate was placed for protection. The samples were placed in an alumina crucible and buried under diamond powders against oxidation. The crucible was transferred to a muffle furnace and heated to ~980 °C in vacuum or argon atmosphere at a rate of 15 °C/min, held at the target temperature for 2 - 24 h and then cooled down to room temperature in furnace. Afterwards, samples were cleaned by sonication in ethanol for 30 min to remove the residual diamond nanoparticle powders and other contaminants. The top surface was ground with sandpapers from 100 grit to 7000 grit, removing 30 - 60 μm surface material. After the fine grinding on 7000 grit sandpapers, the sample surface was deeply etched using 4% (vol. %) nital, to expose the buried DNPs.

### Extraction of DNPs inside bulk samples

As schematically shown in Supplementary Fig. 5a, the thoroughly cleaned and deeply etched sample was dissolved in hydrochloric acid (20 vol.% HCl in ethanol) inside a sealed beaker to avoid contaminations. The hydrochloric acid was excessive to ensure that the sample completely reacted with HCl ($Fe + 2HCl \rightarrow FeCl_2 + H_2(g)$, Fe or $Fe_xC$ dissolved into the acid turned into $FeCl_2$). The remnants were separated from the solution using centrifugation at 5590 × g for 60 min, and washed in ethanol thoroughly ($FeCl_2$ was washed away). Then the cleaned remnant powders were sonicated in ethanol for 20 min.

Finally, the solution containing remnants was drop-deposited onto the TEM grid for characterizations.

### Ab initio simulations

DFT calculations were carried out with Vienna ab initio Simulation Package (VASP)[44]. The Perdew-Burke-Ernzerhof (PBE) functional[45] and the projector augmented wave (PAW) pseudopotentials[46] were used with the energy cutoff of 600 eV. The Fe-diamond interface model was built based on the HRTEM images shown in Fig. 3e and Supplementary Fig. 6. Both the atomic positions and cell parameters were fully relaxed. For comparison, a Fe-graphite model with the same number of atoms was built. The chemical bonding analysis were performed using the crystal orbital Hamilton population (COHP) method[47] as implemented in the Local Orbital Basis Suite Towards Electronic-Structure Reconstruction (LOBSTER) code[48]. Climbing image nudged elastic band calculations (CI-NEB)[49] were based on a supercell with 81 Fe and 192 C atoms, as shown in Supplementary Fig. 7c inset. The stacking sequence of the graphite layers was kept as the conventional a-b stacking to mimic the configuration in experiments. For NEB calculations, the initial and final atomic configurations were relaxed with DFT firstly, which allowed the Fe atom to find an energetically favorable position locally. As shown in Supplementary Fig. 7c, the Fe atom sits on the top of a carbon hexagon for both the initial and final atomic configurations. Therefore, the migration pathway for the Fe atom in the graphite channels would be the continuous hopping between the energetically favorable positions as illustrated in Fig. 4c. The energy saddle point of one NEB pathway is approximately half way between the adjacent two energetically favorable positions (Supplementary Fig. 7c).

### ToF-SIMS analysis

The time-of-flight secondary-ion mass spectroscopy measurements were performed by a ToF-SIMS instrument (IONTOF M6, GmbH, Muenster, Germany) in a negative mode. The data were acquired in frames. Each data frame was generated after 10 scans over the analysis area with one single shot per each pixel. One keV sputter $Cs^+$ ion source with the beam current of 100 nA was used for sample etching in the non-interlaced mode to enhance the yield of negative secondary ions (from C and Fe). The analysis was performed by rastering the primary ion beam (30 keV $Bi^+$, 1 pA) randomly in 128 × 128 pixels over 50.78 μm × 50.78 μm and 4.86 μm × 4.86 μm sample areas, respectively. In total, 6104 scans were acquired on the test specimen and 100 single scans were used for the lateral shift correction over the whole analyzed area. 3D-compositional analysis was implemented in ToF-SIMS by employing sputter ions to remove the uppermost layer of the sample after each analysis scan. This method allows for depth analyses to be conducted.

### MC simulation

The MC simulation used a simple-cubic supercell with the dimension of 50*50*100, and the atomic jump frequency is taken to be ~1 ps. A DNP with the size of d = 5 nm with two surface atomic layers of graphite was placed inside the iron, mimicking the scenario in our experiment. As calculated via NEB, the migration barrier of an isolated Fe atom diffusing in-between graphite layers, $\Delta E_{Fe\text{-}in\text{-}graphite}$, is 0.69 eV for the first van der Waals gap, and 0.28 eV for the next gap (see Fig. 4c and Supplementary Fig. 7c). The model in Fig. 4d used the latter value, because this constitutes the easiest channel available between the graphite (0002) planes for the diffusion of Fe, and the activation energy of 0.28 eV allows modeling the diffusion directly using MC on computer simulation time scale. The chemical potential gradient, $\nabla\mu$ (eV per interatomic distance), biases the direction of Fe flux, to go from the location underneath the diamond particle (higher chemical potential) toward the location above (lower potential). This modifies the migration barrier of Fe atoms in-between the graphite layers to

$0.28 + 0.5 \cdot \nabla\mu$ (moving downward) and $0.28 - 0.5 \cdot \nabla\mu$ (moving upward). Also, the process of Fe atoms entering the graphite layers is not rate limiting, given the graphite layers are defective such that the energy barrier is considerably lower than 0.28 eV. For each MC step, we only consider the DNP as well as the Fe atoms jumping along the channel in-between the two graphite atomic planes. The jump acceptance probability was based on the Metropolis algorithm. We indeed observe the downward translational motion of the DNP. The motion distances as a function of time under various $\nabla\mu$ are shown in Fig. 4d. We did not directly simulate the case of Fe diffusing via the first gap of graphite planes, because the corresponding energy barrier of 0.69 eV would make the MC simulation too computationally expensive. Instead, we choose to use extrapolation to estimate the DNP velocity at 0.69 eV from the results obtained for a range of lower energy barriers (see Supplementary Fig. 10a).

## Data availability
All data generated or analyzed in this study are provided in the article and the Supplementary Information files, and are available from the corresponding authors upon request.

## Code availability
The simulation codes that support the findings of the study are available from the corresponding authors upon request.

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

## Acknowledgements
Y.W. acknowledges supports from the National Key Research and Development Program of China (2022YFB3203600) and K.C. Wong Education Foundation. Z.S. acknowledges the support from National Natural Science Foundation of China (52272162). Y.W. gratefully thanks Prof. Yuefei Zhang's research group at Beijing University of Technology for the help in high-temperature in situ SEM tests, Danli Zhang at Xi'an Jiaotong University (XJTU) for the help in mechanical tests, as well as Hiroaki Matsumoto and Chaobin Zeng from Hitachi High-Tech Company for their help with in situ STEM experiments. S.Z. thanks Chris Yang from Thermo Fisher Scientific Shanghai NanoPort for her technical support on the in situ heating experiments. B.L. thanks You Liu at School of Physics of XJTU for her help in Raman characterization. J.D., W.Z. and E.M. acknowledge XJTU for hosting their work at the Center for Alloy Innovation and Design (CAID). The authors are also indebted to the support by the High-Performance Computing (HPC) platform of XJTU and the International Joint Laboratory for Micro/Nano Manufacturing and Measurement Technologies of XJTU.

## Author contributions
Z.S. conceived the project and supervised it with E.M. Y.W. carried out the experimental investigations with assistance from B.L., L.H., Z.L., Z.D. and C.J.; J.D. conducted the MC simulations; J.L. initially proposed the mechanism to be analogous to Kirkendall effect. Insight into the thermodynamics and kinetic pathway were later provided by Y.W. and E.M. using CALPHAD data and by X.W. and W.Z. using DFT calculations. The data presentation was designed by Y.W. and Z.S., with assistance from L.Z., S.Z., R.Z., Z.D., W.X. and C.F. The writing of the paper was led by E.M. and Y.W. All the authors contributed to the discussions.

## Competing interests
The authors declare no competing interests.
