## [Peer Review File · Nature Communications]

Inward motion of diamond nanoparticles inside an iron crystalREVIEWER COMMENTS

Reviewer #1 (Remarks to the Author):

The observed rigid body motion of diamond particles into bulk iron is indeed very interesting. However the presented theory is not convincing to me and lacks experimental evidence. A carbon concentration gradient and resulting chemical potential gradient is given as the origin of the driving force. This could be supported by measurements of the carbon concentration gradient (for example using SIMS). The chemical potential gradient applied in the modeling is extreme, by many orders of magnitude. The authors claim that the particle trajectories can have millimeter length. Along these distances the concentration (and chemical potential) gradient would be very shallow. Maybe a rough estimate of the resulting migration speed would be helpful here. I'm also skeptical that the solution of carbon from the DNPs would allow for millimeter travel. The supplemental videos look great, but are not conclusive either (is the particle really entering the foil, or migrating along the surface?).

I wonder if the motion of the particle leaves behind a disordered channel that could promote fast diffusion of iron atoms toward the surface. Either way, something has to break the symmetry and help the particle sustain the inward direction.

Further experimental evidence is needed in my opinion before this interesting result can be published.

Reviewer #2 (Remarks to the Author):

The authors report an abnormal and remarkable mass transfer way in solids, i.e., robust diamond nanoparticles entering into solid iron/steel crystals as a whole rather than decomposed individual atoms in the absence of externally applied mechanical force. The preservation of crystalline diamond structure inside iron is beyond our conventional wisdom that diamond is easily graphitized when comes into contact with iron, especially at elevated temperatures the inevitable chemical wear of diamond tools in cutting of steels. The underlying mechanism (involves dissolution of oxide scales on iron, surface transformation from diamond into graphite, and the chemical-potential gradients induced local stress) has been unraveled, using in-situ microscopy and bulk experiments as well as the modelling techniques. The thermodynamic and kinetic processes of the sinking-in and translation of diamond particles in iron have been taken into consideration and analysed by the authors in a very thorough manner, leading to a consistent explanation of their observations. Besides the scientific significance, this work provides a new method for nanodiamonds-dispersion strengthened steels with ultrahigh hardness and wear resistance, superior to what can be currently offered by the familiar powder metallurgy.

To give this work more implications for the developemnt of diamond-based materials and devices, I would like to suggest the authors give a little bit more discussion to clarify the difference between this work and previous studies on diamond etching using other metals such as Ni.

Reviewer #3 (Remarks to the Author):

The Authors demonstrated that diamond nanoparticles (DNP) spread on the surface of solid iron or steel penetrate into the metal to the depths reaching several millimeters at the temperatures close to 1000 C. The sinking of DNPs into iron was discussed in terms of combination of Fe-catalyzed graphitization of diamond, gradient of chemical potential of iron, and fast diffusion of Fe atoms along the graphitized Fe-diamond interface. The phenomenon uncovered in this work is of potential interest for a number of engineering applications. However, the work is not put in a proper perspective with respect to existing literature, and cannot be published in Nature Communications in its present form: 1. The initial sinking of DNPs into subsurface region of iron is related to the good wetting of graphitized diamond by Fe. Similar observations have been reported in the past [1]. This initial sinking has nothing to do with the "Gibbs adsorption isotherm", as claimed in page 5 of the manuscript.

2. Also, it is well-known that small inclusions can move inside the solids driven by the gradient of chemical potential. The early literature on this topic is summarized by Geguzin and Krivoglaz [2]. In this respect, the atomistic Monte Carlo simulations presented in Fig. 4 just illustrate a well-known fact, and their added value is quite limited, since the employed driving forces for migration are many orders of magnitude higher than the ones relevant for the experiment. Naturally, such high driving forces result in high migration rates of several cm per sec. In fact, an exact analytical expression for the migration rate of a cubic particle can be derived, provided that the self-diffusion coefficient of Fe along the Fe-DNP interface is known [3]. The Authors should estimate whether the DNP of 100 nm in size can move with the velocities observed in the experiment under the action of chemical potential gradient caused by variations of carbon content.

3. Following the previous comment, the gradient of carbon content in the near-surface layer is essential for the mechanism of DNPs migration proposed by the Authors. The idea seems disputable to me, since each DNP by itself is a source of carbon, which should level off any externally imposed concentration gradient. Therefore, the Authors should provide an experimental proof of varying carbon concentration in the near-surface layer. Since the quantification of carbon content with EDS in TEM may be challenging, the atomic probe tomography (APT) seems to be better suited for this task.

4. The Authors have to discuss alternative mechanisms of DNPs sinking into iron. For example, it is well-known that small particles can be dragged by moving grain boundaries [4], so that the recrystallization and grain growth in the iron sample may promote the spreading of DNPs into the sample interior.

Minor comment: in page 3, the sentence beginning with "Raman spectroscopy scans..." is repeated twice.

References:

1. C.G. Zimmermann et al., Phys Rev B 64 (2001) 085419.
2. Ya. E. Geguzin, M.A. Krivoglaz, Migration of macroscopic inclusions in solids, Springer, 1973.
3. Z. Suo, Motion of microscopic surfaces in materials, Adv. in appl. mech. 33 (1997) 193.
4. M.F. Ashby, M.R. A. Centamore, Acta metal. 16 (1968) 1081.

Reviewer #1 (Remarks to the Author):

Comment 1

The observed rigid body motion of diamond particles into bulk iron is indeed very interesting. However the presented theory is not convincing to me and lacks experimental evidence. A carbon concentration gradient and resulting chemical potential gradient is given as the origin of the driving force. This could be supported by measurements of the carbon concentration gradient (for example using SIMS).

Reply:

We are glad that the reviewer considers our finding is very interesting. We also highly appreciate the valuable suggestions given by the reviewer. A carbon concentration gradient from the surface towards the interior of bulk Fe has been evidenced by the depth profiles and three-dimensional-compositional images of C and Fe from the time-of-flight secondary-ion mass spectroscopy analysis (ToF-SIMS, Fig. 4b).

Fig. 4b ToF-SIMS analysis of a sample heated in furnace for 1 hour (all dispersed DNPs have entered the iron matrix) and then quenched in water to retain the carbon distribution at high temperatures. The ToF-SIMS spectra showing the depth profiles of secondary ions of C⁻ (grey) and Fe⁻ (blue) in the sputtered volume from the top surface to a ~60 μm depth over an area of ~50×50 μm². The corresponding 3D images of the depth profiles visualizing the opposite concentration gradients of C and Fe. The inset ToF-SIMS spectrum displays the depth profile of C⁻ in a much smaller volume of the sample (the analyzed area is ~5×5 μm² and the depth is ~500 nm).

Action taken

We have added the SIMS characterization in Fig. 4b.

Comment 2

The chemical potential gradient applied in the modeling is extreme, by many orders of magnitude. The authors claim that the particle trajectories can have millimeter length.

Along these distances the concentration (and chemical potential) gradient would be very shallow. I'm also skeptical that the solution of carbon from the DNPs would allow for millimeter travel. Maybe a rough estimate of the resulting migration speed would be helpful here.

Reply:

Diamond nanoparticles at the millimeter-depth of the Fe matrix have indeed been found (verified by the Raman spectroscopy, Fig. 3c), though the signal intensity of diamond decreases significantly as the depth increases (only some leading DNPs with high enough local concentration gradient can reach the millimeter depth). In the following, we make an explanation about how the local concentration gradient is built up and sustained.

Here the source of C is the graphitized surface layer of each and every DNP gradually dissolving to satisfy the solubility in the surrounding Fe. The carbon concentration profile could be maintained for a long time via the continual supply from these DNPs (the number of DNPs entering Fe is sufficiently large). Note that different from the conventional carburizing, in which the carbon source only exists outside the steel and the carbon concentration gradient becomes shallower and shallower along the millimeter depth direction, here each engulfed DNP with a gradually graphitized and meanwhile dissolving interfacial layer acts as a movable source of carbon by itself. As a result, the large carbon concentration gradient extends forward along with the moving DNPs (inhomogeneous concentration field). The driving force for the inward motion of each DNP is the local chemical potential gradient across the nanoparticle itself instead of across the long millimeter-distance (we have redrawn the schematic diagram in Fig. 4a to highlight this). Take the leading DNP as an example, as it moves forward deeper and deeper, it keeps encountering iron interior with less carbon solute content (i.e., the DNP always sees a higher Fe concentration at the location in front of its moving trajectory). The dissolved carbon atoms at the location underneath the DNP diffuse into the "virgin" iron more rapidly than into the already-carbon-enriched locations above the DNP. That is, in its surroundings the DNP is always accompanied by a local concentration gradient, see the C profile in a local (smaller volume) region, likely around some DNPs, as displayed in the inset of the TOF-SIMS spectra, Fig. 4b. As a result, the carbon concentration gradient extends forward along with the moving DNPs (inhomogeneous concentration field), and makes the millimeter travel of DNPs possible.

We also have performed a back-of-the-envelope numerical estimate of the travel velocity of the DNP. The DNP is treated as a sphere with diameter d , and the Fe

volume flux J_{cV} (in the unit of m^3/s) through the equatorial plane of the diamond sphere can be written as

(2)

where dc/dz is the Fe concentration gradient (or the negative carbon concentration gradient, in units of atoms/ m^3/m) across the DNP, D_{Fe} is the effective diffusivity of Fe atoms at the Fe-DNP interface, δ_s is the thickness of Fe atoms flux and equals the diameter of a Fe atom, about 0.25 nm. The translational movement of the DNP fills in the space left by leaving Fe atoms, and the volume flux of iron atoms J_V can also be written as

(3)

where the translation velocity v can be expressed as:

(4)

where $D_{Fe} = 6 \times 10^{-11} m^2/s$ (see the Supplementary discussion II and III for details). For a DNP with $d = 100$ nm, with the maximum chemical concentration difference across its two poles (at 1,250 K, the maximum carbon concentration reaches 1.8 wt. %, i.e., close to 7 at. %, see Fig. S8), Equation (4) predicts a maximum $v \sim 120$ nm/s. This estimated DNP velocity is comparable with that observed in experiments (the DNPs reach up to ~ 1 mm within 5 h, the maximum motion velocity is hence ~ 0.2 mm/h or ~ 50 nm/s).

Action taken

We have added the discussion about the local concentration gradient in the main text (Page 6), and highlighted it in Fig. 4a. We also have supplemented the back-of-the-envelope numerical estimate of the velocity of DNP in Supporting Information.

Comment 3

The supplemental videos look great, but are not conclusive either (is the particle really entering the foil, or migrating along the surface?). I wonder if the motion of the particle leaves behind a disordered channel that could promote fast diffusion of iron atoms toward the surface. Either way, something has to break the symmetry and help the particle sustain the inward direction. Further experimental evidence is needed in my opinion before this interesting result can be published.

Reply:

Diamond nanoparticles (DNPs) move into the Fe foil, rather than migrate along the

surface. Please see the real-time observations of the sink-in process of DNPs recorded in an added video (Supplementary Video 2), where the yet non-entered particles on the surface of Fe serve as the reference in a fixed coordinate system. The right panel of this video demonstrates the real-time sink-in process of diamond nanoparticles under the secondary-electron imaging, which is simultaneously recorded with the scanning-TEM observations (the left panel).

We agree with the referee that the motion of the particle may leave behind a “disordered channel” that could promote fast diffusion of iron atoms toward the surface. To verify this, we have conducted an in-situ TEM experiment to trace the motion of the DNP cluster (marked in orange, Fig. R1a). After the DNP entered the Fe matrix, we stopped heating and then further thinned the sample to expose the engulfed DNP, as shown in the TEM image (Fig. R1b), from which we can see the contrast of defects (distortion of the crystal lattice) in the upper zone of the DNP indicating a possible easy channel left behind by the downward DNP for the Fe atoms diffusion at high temperatures.

Fig. R1 (a) Snapshots from the in-situ TEM video showing the downward motion of DNPs into the Fe matrix; (b) The TEM image of the thinned sample after in-situ heating experiment, showing the engulfed DNP inside the Fe matrix.

Action taken

We have added this discussion on page 7: “*Note that the downward motion of the DNP may leave behind an “easy channel” that could further promote the fast upward diffusion of the Fe flux towards the sample surface. Fe atoms leaving from the upper interface of Fe/DNP help to sustain the local concentration gradient around the DNP and hence continuous motion of the particle.*”

Reviewer #2 (Remarks to the Author):

Comment 1

The authors report an abnormal and remarkable mass transfer way in solids, i.e., robust diamond nanoparticles entering into solid iron/steel crystals as whole rather than decomposed individual atoms in the absence of externally applied mechanical force. The preservation of crystalline diamond structure inside iron is beyond our conventional wisdom that diamond is easily graphitized when comes into contact with iron, especially at elevated temperatures the inevitable chemical wear of diamond tools in cutting of steels. The underlying mechanism (involves dissolution of oxide scales on iron, surface transformation from diamond into graphite, and the chemical-potential gradients induced local stress) has been unraveled, using in-situ microscopy and bulk experiments as well as the modelling techniques. The thermodynamic and kinetic processes of the sinking-in and translation of diamond particles in iron have been taken into consideration and analysed by the authors in a very thorough manner, leading to a consistent explanation of their observations. Besides the scientific significance, this work provides a new method for nanodiamonds-dispersion strengthened steels with ultrahigh hardness and wear resistance, superior to what can be currently offered by the familiar powder metallurgy.

Reply:

We thank the reviewer for pointing out the significance of our work.

Comment 2

To give this work more implications for the development of diamond-based materials and devices, I would like to suggest the authors give a little bit more discussion to clarify the difference between this work and previous studies on diamond etching using other metals such as Ni.

Reply:

It has been known that diamond can be etched by Fe, Ni and Co etc. via the metal-catalyst-assisted graphitization of diamond and the carbon dissolution in metals. In our work, the Fe-catalyzed surface graphitization of DNPs is utilized for the formation of the easy diffusion channel at the Fe-diamond interface, and the gradually dissolved carbon atoms from the graphite surface layers sets up the chemical concentration gradient as the driving force for the rigid motion of DNPs. Our additional experiments (see Fig. R2) have shown that similar motion of DNPs as whole can occur inside the Ni crystals as well.

The introduction of DNPs into Fe or Ni matrix here is very different from the Fe- or Ni- catalyzed diamond etching for patterning or creating nanopores in an H₂ atmosphere

at high temperatures. The reported diamond etching mechanism mainly involves carbon dissolution in metal Fe, Ni etc., diffusional transport to the metal-gas interface and carbon desorption in the form of methane. (*Ralchenko, et al. Catalytic interaction of Fe, Ni and Pt with diamond films: patterning applications. Diamond & Related Materials 2.5-7(1993):904-909*; *Mehedi et al. Etching mechanism of diamond by Ni nanoparticles for fabrication of nanopores, Carbon 59(2013)448-456*; *Catalytic interaction of Fe, Ni and Pt with diamond films: patterning applications*).

Fig. R2. The representative Raman spectrum acquired at the ~50 μm depth of the Ni-DNPs sample after the top surface ground and etched away. The inset SEM image showing the exposed DNPs after the Ni matrix was deeply etched using acid solution.

Reviewer #3 (Remarks to the Author):

Comment 1

.....The Authors demonstrated that diamond nanoparticles (DNP) spread on the surface of solid iron or steel penetrate into the metal to the depths reaching several millimeters at the temperatures close to 1000 C. The sinking of DNPs into iron was discussed in terms of combination of Fe-catalyzed graphitization of diamond, gradient of chemical potential of iron, and fast diffusion of Fe atoms along the graphitized Fe-diamond interface. The phenomenon uncovered in this work is of potential interest for a number of engineering applications. However, the work is not put in a proper perspective with respect to existing literature, and cannot be published in Nature

Communications in its present form.

Reply:

We really appreciate the referee's valuable suggestions and the recommended literature, which are very helpful for us to solidify the underlying mechanism. We have added experiments, analytical calculations, and revised the manuscript according to the referee's suggestions and cited the relevant work mentioned by the referee. In the following, we provide a point-to-point response to address the review questions.

Comment 2

The initial sinking of DNPs into subsurface region of iron is related to the good wetting of graphitized diamond by Fe. Similar observations have been reported in the past [1]. This initial sinking has nothing to do with the "Gibbs adsorption isotherm", as claimed in page 5 of the manuscript.

References:

1. C.G. Zimmermann et al., Phys Rev B 64 (2001) 085419.

Reply:

We agree with the referee that the Fe flux spreads over the particles via fast surface diffusion, resulting in a high degree of wetting. The capillary force arises from the interaction between the Fe flux and the DNP surface, directed along the nanoparticle/Fe interface. It forces the mass flow during the initial sinking-in process. According to the model proposed by Zimmermann et al. [Zimmermann, C. G., et al. *Burrowing of Co Nanoparticles on Clean Cu and Ag Surfaces. Phys. Rev. Lett.* 83.6(1999):1163-1166; Zimmermann, C. G., et al. *Burrowing of nanoparticles on clean metal substrates: Surface smoothing on a nanoscale. Phys. Rev. B* 64.8(2001):085419], we have estimated the capillary stress, which can reach the gigapascal-level, high enough to "drag" DNPs in to the Fe matrix.

We have removed the discussion about "Gibbs adsorption isotherm", cited the work of Zimmermann et al, and rewritten this part on Page 5: "*Freshly exposed Fe atoms flow from underneath the DNP and wrap around it via fast surface diffusion (see Supplementary Discussions Section I). This action can be construed as Fe striving to cover up or wet carbon to lower the surface energy of the DNP. A capillary force arises*

from the interaction between the Fe flux and the DNP surface, directed along the DNP-Fe interface, driving the particle towards the inside. The resultant stress at the bottom interface of Fe-DNP can reach gigapascal level, which is estimated according to the burrowing model proposed by Zimmermann et al”.

Comment 3

It is well-known that small inclusions can move inside the solids driven by the gradient of chemical potential. Geguzin and Krivoglaz [2] summarize the early literature on this topic. In this respect, the atomistic Monte Carlo simulations presented in Fig. 4 just illustrate a well-known fact, and their added value is quite limited, since the employed driving forces for migration are many orders of magnitude higher than the ones relevant for the experiment. Naturally, such high driving forces result in high migration rates of several cm per sec. In fact, an exact analytical expression for the migration rate of a cubic particle can be derived, provided that the self-diffusion coefficient of Fe along the Fe-DNP interface is known [3]. The Authors should estimate whether the DNP of 100 nm in size can move with the velocities observed in the experiment under the action of chemical potential gradient caused by variations of carbon content.

References:

2. Ya. E. Geguzin, M.A. Krivoglaz, Migration of macroscopic inclusions in solids, Springer, 1973.
3. Z. Suo, Motion of microscopic surfaces in materials, Adv. in appl. mech. 33 (1997)

Reply:

We thank the referee for the recommendation of the literature we omitted. We have cited the earlier literature at the discussion section about the thermodynamic driving force for the motion of DNPs: “*We first show that there is a thermodynamic driving force available to sustain the upward mass transport of Fe. It has been reported that small inclusions can move inside the solids driven by the gradient of chemical potential²¹*” (on Page 5 in the main text).

The work summarized by Geguzin and Krivoglaz about the migration of inclusions inside solids under the chemical potential gradient (Fig. 44~Fig. 51 in Reference 2 “Ya. E. Geguzin, M.A. Krivoglaz, Springer, 1973”) are more like the well-known Kirkendall effect, in which a rigid-body mechanical motion due to the chemical potential driven lattice vacancy exchange/accumulation. Our case here has some differences. First, instead of chemically inert solid particles (or gas bubbles) as fiducial markers, here our DNP “fiducial markers” are actively dissolving, thus creating the concentration gradient in Fe crystals. Second, instead of the lattice vacancy exchange/accumulation

mechanism as in the Kirkendall experiment, here it is the interfacial diffusion of Fe that generates local stresses to induce the rigid translation of particles. The Fe concentration gradient in our case is set up by carbon dissolution, driving the interfacial-diffusional flow of Fe, and subsequently the rigid translational motion of the DNP “markers”.

The Monte Carlo simulation performed here is mainly to show that the DNP can move downward inside the Fe matrix, only driven by the chemical potential gradient $\nabla\mu$ (for example $\nabla\mu= 0.005$ eV), and the motion velocity is proportional to $\nabla\mu$. In experiment, the equilibrium solubility (7% at. %) of C in FCC Fe at high temperature (see Fig. S8) corresponds to the maximum chemical potential gradient $\nabla\mu$ of ~ 0.005 eV/atom across the two poles of a DNP $d = 100$ nm in diameter (for smaller DNPs, the value of $\nabla\mu$ could be higher). It should be noted that the direct driving force for the inward motion of each DNP is the local chemical potential gradient across the nanoparticle itself instead of across the long millimeter-distance (please see the explanation in our reply to Comment 4). As for the higher chemical potential gradients $\nabla\mu$ used in the Monte Carlo simulation, we intended to show the dependence of the velocity on different chemical potential gradients. The extrapolation based on the Monte Carlo simulation suggests a v of ~ 25 nm/s for a DNP $d = 100$ nm in diameter under the maximum chemical potential gradient (~ 0.005 eV/atom across its two poles), comparable with the maximum experimental observation of ~ 50 nm/s (the DNPs reach up to ~ 1 mm within 5 h, the maximum motion velocity is hence ~ 0.2 mm/h or ~ 50 nm/s). Please see Supplementary discussion II and III in Supporting Information for details. In addition to the extrapolation from the direct MC simulations of DNP motion discussed above, we have also performed a back-of-the-envelope numerical estimate of the travel velocity of the DNP. The DNP is treated as a sphere with diameter d , and the Fe volume flux J_{cV} (in the unit of m^3/s) through the equatorial plane of the diamond sphere can be written as

(2)

where dc/dz is the Fe concentration gradient (or the negative carbon concentration gradient, in units of atoms/ m^3/m) across the DNP, D_{Fe} is the effective diffusivity of Fe atoms at the Fe-DNP interface, δ_s is the thickness of Fe atoms flux and equals the diameter of a Fe atom, about 0.25 nm. The translational movement of the DNP fills in the space left by leaving Fe atoms, and the volume flux of iron atoms J_V can also be written as

(3)

where the translation velocity v can be expressed as:

(4)

where $D = 6 \times 10^{-11} \text{ m}^2/\text{s}$ (this value is not available in experiments, and we performed a two-dimensional MC simulation to calculate it. Please see Supplementary Discussion III and Fig. S11 for details). For a DNP with $d = 100 \text{ nm}$, with the maximum chemical concentration difference across its two poles (at 1,250 K, the maximum carbon concentration reaches 1.8 wt. %, i.e., close to 7 at. %, see Fig. S8), Equation (6) predicts a maximum $v \sim 120 \text{ nm/s}$. This estimated DNP velocity is close to what was observed in experiments $\sim 50 \text{ nm/s}$.

The above estimation indicates that a DNP of 100 nm in size could move with the velocities observed in the experiment under the action of the local chemical potential gradient across the DNP. The consistence in the maximum velocity from the experimental observation and the analytical calculations based on the driving force of the carbon concentration gradient also suggests that the large enough chemical potential difference across the DNP can be reached in experiment.

Comment 4

Following the previous comment, the gradient of carbon content in the near-surface layer is essential for the mechanism of DNPs migration proposed by the Authors. The idea seems disputable to me, since each DNP by itself is a source of carbon, which should level off any externally imposed concentration gradient. Therefore, the Authors should provide an experimental proof of varying carbon concentration in the near-surface layer. Since the quantification of carbon content with EDS in TEM may be challenging, the atomic probe tomography (APT) seems to be better suited for this task.

Reply:

We thank the referee for the valuable suggestion.

Firstly, we make an explanation about how the local chemical concentration gradient across individual DNPs is built up and sustained. The source of C is the graphitized surface layer of each and every DNP gradually dissolving. During or even before the initial sink-in of DNPs, the surface graphitization of diamond followed by the carbon dissolution in Fe has occurred. The high enough carbon concentration in the near-surface region could be maintained for a long time via the continual supply from these DNPs (the number of DNPs entering Fe is sufficiently large), and the concentration

gradient is built up in the meantime. As each engulfed DNP with a gradually graphitized and meanwhile dissolving interfacial layer moves, the carbon concentration gradient extends forward along with them (the inhomogeneous concentration field). The driving force for the inward motion of each DNP is the local chemical potential gradient across the nanoparticle itself instead of across the long millimeter-distance (we have redrawn the schematic diagram in Fig. 4a to illustrate this). Take the leading DNP as an example, as it moves forward deeper and deeper, it keeps encountering iron interior with less carbon solute content (i.e., the DNP always sees a higher Fe concentration at the location in front of its moving trajectory). The dissolved carbon atoms at the location underneath the DNP diffuse into the “virgin” iron more rapidly than into the already-carbon-enriched locations above the DNP. That is, in its surroundings the DNP is always accompanied by a local concentration gradient. As a result, the carbon concentration gradient extends forward along with the moving DNPs (inhomogeneous concentration field), and makes the long-distance travel of leading DNPs possible.

A carbon concentration gradient in the near-surface layer has been evidenced by the depth profiles and three-dimensional-compositional images of C and Fe from the time-of-flight secondary-ion mass spectroscopy analysis (ToF-SIMS, Fig. 4b). The ToF-SIMS is a surface sensitive analytical method and can demonstrate the carbon concentration gradient across a much larger distance compared with APT. The C profile in a local (much smaller volume) region, likely comes from the local concentration gradient around some DNPs, as displayed in the inset of the TOF-SIMS spectra, Fig. 4b. We also have tried to characterize the local carbon profile across a single DNP quantitatively, but unfortunately, we didn't make it. We had little chance to include the leading diamond nanoparticles (their number density is quite low) inside a tiny APT needle sample (non-site-specific sample preparation via the electrochemical polishing). As for the reason why we didn't use the site-specific APT specimen preparation (FIB-millings and lift-out) to locate the individual diamond nanoparticles inside Fe matrix, because it has been well-documented that the carbon profile versus depth in Fe matrix or steels can be obviously altered by the high-energy focused ion beam irradiation (Wang, Jing, *et al. Scientific Reports* 7.1 (2017): 15813.; Basa *et al. Metall Mater Trans A* 45, 1189–1198, 2014).

Fig. 4 | Mechanism for the inward motion of DNP into the iron crystal. (a) Schematic depicting the steps involved in the process. (b) ToF-SIMS analysis of a sample heated in furnace for 1 hour (all dispersed DNP have entered the iron matrix) and then quenched in water to retain the carbon distribution at high temperatures as much as possible. The ToF-SIMS spectra showing the depth profiles of secondary ions of C^- (grey) and Fe^- (blue) in the sputtered volume from the top surface to a $\sim 60 \mu m$ depth over an area of $\sim 50 \times 50 \mu m^2$. The corresponding 3D images of the depth profiles visualizing the opposite concentration gradients of C and Fe. The inset ToF-SIMS spectrum displays the depth profile of C^- in a much smaller volume (the analyzed area is $\sim 5 \times 5 \mu m^2$ and the depth is ~ 500 nm).

Comment 5

The Authors have to discuss alternative mechanisms of DNP sinking into iron. For example, it is well-known that small particles can be dragged by moving grain boundaries [4], so that the recrystallization and grain growth in the iron sample may promote the spreading of DNP into the sample interior.

References:

4. M.F. Ashby, M.R. A. Centamore, *Acta metal.* 16 (1968) 1081.

Reply and action taken:

We appreciate the referee's suggestion and the recommendation of the nice work of Ashby et al. As shown in the *Acta Metal* paper, not any small particles are mobile under the dragging of migrating grain boundaries, for example the Al_2O_3 nanoparticles in Cu matrix. Here we have not observed obvious aggregation of DNPs on the grain boundary or DNPs-denuded region near the grain boundary, indicating that the dragging effect of grain boundaries is not dominant for the inward transitional motion of DNPs in Fe matrix. Please see the added Fig. S12 in Supporting Information (as shown below).

We have added some discussions about other alternative mechanisms on Page 9 in the main text: *“The mechanism proposed above is more plausible than other alternatives we can find out at present, and two of them are discussed in the following. First, it has been reported that migrating grain boundaries can drag with them some solid or liquid small oxide particles inside the metal matrix, causing the accumulation of particles on the grain boundary^{31, 32}. In our case, no obvious aggregation of DNPs on the grain boundary or DNPs-denuded region near the grain boundary has been found (see Supplementary Fig. S12). Instead, the DNPs are found spread-out inside the Fe matrix. Second, one may argue that DNPs inside the iron interior come from the dissolution of the original DNPs, followed by the diffusion and then segregation of supersaturated carbon atoms to re-precipitate diamond particles upon cooling. However, this diffusional process relying on individual atoms would be the same as that in the conventional carburizing treatment, which never produced diamond inside the steels. What's more, even with supersaturation and segregation, the precipitation of diamond is far more difficult than the nucleation of the routinely observed graphite and cementite: the minimum temperature/pressure required for diamond formation in the Fe-C system would be about 1200 °C/5 GPa³³”.*

Fig. S12 | Comparison of the deeply etched surface with and without DNPs. a, SEM image of the deeply-etched original sample with a relatively smooth surface. **b**, SEM image of the deeply-etched DNPs-iron sample and the SEM view of two enlarged zones from different grains with different etching depths. Note that these DNPs are not concentrated at grain boundaries of iron and no obvious DNPs-free regions near the grain boundary have been found.

Comment 6

Minor comment: in page 3, the sentence beginning with “Raman spectroscopy scans...” is repeated twice.

Reply:

We thank the referee for pointing out this typo, and we have removed the repetition and double-checked the full text thoroughly. Again, we thank the reviewer for all valuable comments and suggestions, which have helped us in improving the quality of the manuscript.

REVIEWER COMMENTS

Reviewer #1 (Remarks to the Author):

My comments have been adequately addressed.

Reviewer #2 (Remarks to the Author):

The questions from reviewers have been answered. I have no more question for this work. I believe this work is of significance for understanding the properties of diamond.

Reviewer #3 (Remarks to the Author):

The authors have addressed some of my comments, however, several problems with their interpretations of the data are still unresolved:

Concerning the driving force for DNP motion: the equations (2)-(4) in their response letter are fine, yet the driving force for diffusion of Fe atoms cannot be written as dC/dz (gradient of carbon concentration). Such form of Eq. (4) would be fine in the case of substitutional solid solution, whereas Fe-C is an interstitial one. In the bulk of Fe the C atoms diffuse through the interstitial sites, which means the interstitial sublattice should be considered as a third component in the Fe-C alloy. Formally, the Eq. (4) should be modified by activity coefficient $[du_{Fe}/dc]/kT$, which can be calculated using the free energy diagram in Fig. S8 of the Supplementary material. In the Fe-rich region of this diagram the dependence of free energy on composition is close to linear, which means that the activity coefficient is small. For example, in the imaginary case when C atoms perfectly fit into interstitial sites of the Fe lattice, C diffusion would have no effect whatsoever on the Fe matrix.

Regarding the nature of the concentration gradient of C in the proximity of DNP, I find the explanation given by the authors to be rather questionable. They propose that the particle itself generates a concentration gradient of C in its surroundings, driving the particle towards the interior of Fe. This analogy appears reminiscent of Baron Munchausen pulling himself up from the river by his own hair, as depicted in "Baron Munchausen's Narrative of his Marvellous Travels and Campaigns in Russia, etc. [By R. E. Raspe.], Pt. 1, 1786." The interface self-diffusion coefficient of Fe calculated by the authors is quite high, and I expect the C diffusion coefficient to be several orders of magnitude higher. With such high diffusivity the formation of any significant concentration gradient along the DNP of 100 nm in size is highly unlikely. In their ToF-SIMS depth profile the authors do see a macroscopic gradient of C concentration spread over tens of micrometers. This gradient may be caused by density gradient of DNPs across the Fe sample depth. May be this gradient causes stress gradient in Fe which is high enough to drive the DNP migration, but this has to be supported by numerical estimates.

Reviewer #3 (Remarks to the Author):

Comment 1

The authors have addressed some of my comments, however, several problems with their interpretations of the data are still unresolved:

Concerning the driving force for DNP motion: the equations (2)-(4) in their response letter are fine, yet the driving force for diffusion of Fe atoms cannot be written as dC/dz (gradient of carbon concentration). Such form of Eq. (2) would be fine in the case of substitutional solid solution, whereas Fe-C is an interstitial one. In the bulk of Fe, the C atoms diffuse through the interstitial sites, which means the interstitial sublattice should be considered as a third component in the Fe-C alloy. Formally, the Eq. (4) should be modified by activity coefficient $[du_{Fe}/dc]/kT$, which can be calculated using the free energy diagram in Fig. S8 of the Supplementary material. In the Fe-rich region of this diagram the dependence of free energy on composition is close to linear, which means that the activity coefficient is small. For example, in the imaginary case when C atoms perfectly fit into interstitial sites of the Fe lattice, C diffusion would have no effect whatsoever on the Fe matrix.

Reply

We really appreciate all the comments and suggestions from the referee, which are very helpful for us to make the explanations clearer.

We agree with the referee that the concentration gradient of Fe should not be equal to the dC/dz , given the effect of the third component (interstitial sublattice) in Fe-C system. The differential expression in our Eq. (2) and Eq. (4) may be misleading, and we have made modifications. For a crude approximation of the DNP motion velocity driven by the concentration difference, the mass conservation can be used in the DNP local setup. No need to model the process of interstitial diffusion. That is, the downward movement of the DNP fills in the space left by the up transported Fe atoms.

If the DNP is treated as a sphere with diameter d , and the transported Fe mass flux M_{cV} (in the unit of g/s) through the equatorial plane of the diamond sphere can be written as

$$(2)$$

where Δc is the concentration difference of Fe across the DNP; D_{Fe} is the effective diffusivity of Fe atoms at the Fe-DNP interface (graphite layers), here taking the value as $\sim 3 \times 10^{-9}$ m²/s (please see our response to the referee's *Comment 2* and Supplementary Discussion III for details); δ_s is the thickness of Fe atoms flux and equals the diameter of a Fe atom, about 0.25 nm, and Ω (atomic volume of Fe) is 0.0117

nm^3 . The downward movement of the DNP fills in the space left by the up transported Fe atoms, and the mass flux of Fe atoms M_{Fe} can also be written as

(3)

By $M_{\text{Fe-up}}=M_{\text{Fe}}$, the translation velocity v can be expressed as:

(4)

For a DNP with $d=100$ nm and the, even if the Fe concentration difference across its two poles is as small as 0.01 at.%, the estimated velocity of DNP can reach ~ 10 nm/s, which is in the same order as what was observed in experiments, ~ 50 nm/s.

Action taken

We have revised the Supplementary discussion III in Supplementary Material accordingly.

Comment 2

Regarding the nature of the concentration gradient of C in the proximity of DNP, I find the explanation given by the authors to be rather questionable. They propose that the particle itself generates a concentration gradient of C in its surroundings, driving the particle towards the interior of Fe. This analogy appears reminiscent of Baron Munchausen pulling himself up from the river by his own hair, as depicted in "Baron Munchausen's Narrative of his Marvellous Travels and Campaigns in Russia, etc. [By R. E. Raspe.], Pt. 1, 1786." The interface self-diffusion coefficient of Fe calculated by the authors is quite high, and I expect the C diffusion coefficient to be several orders of magnitude higher. With such high diffusivity the formation of any significant concentration gradient along the DNP of 100 nm in size is highly unlikely.

Reply

The analogy is interesting and inspiring. If the DNP is compared to Baron Munchausen, the Fe matrix can be regarded as the mire where Baron Munchausen stuck in. Carbon dissolution into Fe induces a chemical potential gradient in the Fe matrix. That is, it is the difference in Fe chemical potential that drives the movement of Fe, which “pushes” the downward motion of DNP.

Kinetically, the upward flow of Fe along the interface IS FASTER than the downward (interstitial) carbon flux in the Fe lattice for the Fe-C intermixing. This Fe relocation is sufficiently fast because Fe atoms only need to travel a very short distance around the tiny DNP. The migration energy of Fe on graphite is only 0.0068 eV^[1]. In the absence

of such an open graphite surface, the graphitized atomic layers on the DNP surface offer an easy migration channel for Fe diffusion, as shown in Fig. R1. Our Nudged Elastic Band (NEB) DFT calculations (Fig. R1d, also see Fig. S7 and Methods section for details) suggest that the energy barrier for Fe migration along the interfacial graphite is 0.69 eV for the first gap with the outmost (0002) atomic plane bonded with Fe, and 0.28 eV for the next gap in between the neighboring (0002) layers (see Fig. R1c below). Such activation barriers are much lower than the ~ 1.5 eV^[2] of the interstitial diffusion of C in Fe lattice, or the ~ 3 eV for self-diffusion of Fe atoms^[2]. Therefore, the easiest kinetic path for Fe diffusion is the gap in between the neighbouring (0002) layers with an interplanar spacing of ~ 0.34 nm, which offers extra space for the migrating Fe atoms (0.25 nm in diameter). Note that the graphite layers are not perfect crystals but with many flaws and defects which provide enough sites for Fe atoms to enter/leave graphite layers (as shown in the HRTEM image in Fig. R1a,b).

In this scenario, the C diffusion coefficient is much lower than Fe, rather than orders of magnitude higher as assumed by the referee. The calculated diffusion coefficient of Fe atoms in the graphite layers is 3×10^{-9} m²/s (for the 0.28 eV diffusion barrier), whereas the interstitial diffusion coefficient of C atoms in Fe at the same temperature (~ 1200 K) is $\sim 8 \times 10^{-12}$ m²/s^[3]. Such a difference in diffusivity makes it possible for the formation of a significant concentration gradient along the DNP of 100 nm in size.

Fig. R1. **a.** The TEM image of the graphite layer (fast diffusion channel) on DNP. The upper right inset is the EELS spectra taken from the surface layer, verifying the SP²-hybridized carbon. **b.** The enlarged view of the graphite layers on DNP surface. **c.** The schematic showing the upward diffusion of Fe atoms in the graphite layers on the leading DNP. **d.** The migration path and energy barrier predicted by NEB analysis, for the diffusional hopping of Fe atom along the channel in-between graphite (0002) planes. The energy barrier for Fe migration is 0.69 eV in the first gap with one (0002) atomic plane bonded with Fe, and 0.28 eV for the next gap in between the neighboring (0002) layers.

Action taken

We have revised Fig. 4 to highlight the Fe atoms upward diffusion in the easy channel of graphite layers, and the main text has been modified accordingly.

Comment 3

In their ToF-SIMS depth profile the authors do see a macroscopic gradient of C concentration spread over tens of micrometers. This gradient may be caused by density gradient of DNPs across the Fe sample depth. Maybe this gradient causes stress gradient in Fe which is high enough to drive the DNP migration, but this has to be supported by numerical estimates.

Reply

The density gradient of DNPs may contribute to the macroscopic gradient of carbon

concentration measured by the TOM-SIMS. We have estimated this contribution from the undissolved DNPs at different depths in the $\sim 50 \times 50 \mu\text{m}^2$ scanned area. According to our statistic analysis, the SIMS scan near the surface sees 23 DNPs per μm^2 in the scanned area, and the scan at the maximum depth probed in the same scanned area sees 19 DNPs per μm^2 (see Fig. R2). The C concentration decrease resulting from this difference in DNPs is only $\sim 17\%$, which is lower than the total C concentration variation ($\sim 50\%$) measured by the TOM-SIMS.

Fig. R2. SEM images showing DNPs dispersion at different depths (a) $\sim 1 \mu\text{m}$ and (b) $\sim 60 \mu\text{m}$, respectively corresponding to the near surface and maximum depths probed by SIMS. The inset SEM images is the enlarged views of the DNPs dispersed in Fe matrix.

References

- [1] Boukhvalov, D. W. First-principles modeling of the interactions of iron impurities with graphene and graphite. *Phys. Status Solidi B* 248, 1347-1351 (2011).
- [2] Gale, W. F., Totemeier, T. C. *Smithells metals reference book* (Eighth edition). Vol. 13 (Elsevier, 2003).
- [3] Timmerscheidt, T. A., Von Appen, J., & Dronskowski, R. A molecular-dynamics study on carbon diffusion in face-centered cubic iron. *Computational Materials Science*, 91, 235-239 (2014).

REVIEWER COMMENTS

Reviewer #3 (Remarks to the Author):

There is still room for improvement of the revised manuscript. As I wrote in one of my previous comments, the driving force for Fe diffusion flux around DNP cannot be written as $\Delta c/d$; it should be $-du_{\text{Fe}}/dz$, where u_{Fe} is the chemical potential of Fe in the Fe-C solid solution. The difference between the two can reach several orders of magnitude. More details about thermodynamics of interstitial solid solutions can be found in M. Hillert, L.I. Staffansson, *Acta chemica Scandinavica* 24 (1970) 3618.

The authors have all tools to calculate this gradient (the Gibbs diagram of the Fe-C solution, and the knowledge of C depth gradient), and it is a pity they have not done it.

Reviewer #3 (Remarks to the Author):

Comment

There is still room for improvement of the revised manuscript. As I wrote in one of my previous comments, the driving force for Fe diffusion flux around DNP cannot be written as $\Delta c/d$; it should be $-du_{\text{Fe}}/dz$, where u_{Fe} is the chemical potential of Fe in the Fe-C solid solution. The difference between the two can reach several orders of magnitude. More details about thermodynamics of interstitial solid solutions can be found in M. Hillert, L.I. Staffansson, *Acta chemica Scandinavica* 24 (1970) 3618.

The authors have all tools to calculate this gradient (the Gibbs diagram of the Fe-C solution, and the knowledge of C depth gradient), and it is a pity they have not done it.

Reply

We thank the referee for the comment and recommendation of the literature. According to the referee's suggestion, we have modified the Eq.(2) and Eq.(4) by the activity coefficient, considering the effect of the interstitial sublattice in non-ideal Fe-C solid solution system. In the Fe-rich region, the activity coefficient can be expressed as $(d\mu_{\text{Fe}}/dc)/k_{\text{B}}T$, and the calculated value using the Gibbs energy diagram in Fig. S8 is in the range of 0.1~ 0.3.

For a crude approximation of the DNP motion velocity driven by the chemical potential gradient, the mass conservation can be used in the DNP local setup. If the DNP is treated as a sphere with diameter d , and the transported Fe mass flux $M_{\text{Fe-up}}$ (in the unit of g/s) through the equatorial plane of the diamond sphere can be written as

(2)

where D_{Fe} is the effective diffusivity of Fe atoms at the Fe-DNP interface (graphite layers), here taking the value as $\sim 3 \times 10^{-9} \text{ m}^2/\text{s}$; ∇c is the Fe concentration gradient; δ_{s} is the thickness of Fe atoms flux and equals the diameter of a Fe atom, about 0.25 nm, and Ω (atomic volume of Fe) is 0.0117 nm^3 . The downward movement of the DNP fills in the space left by the up transported Fe atoms, and the mass flux of Fe atoms M_{Fe} can also be written as

(3)

By $M_{\text{Fe-up}}=M_{\text{Fe}}$, the translation velocity v can be expressed as:

(4)

where Δc is the concentration difference of Fe across the DNP with diameter of d . For a DNP with $d=100$ nm, even if the Fe concentration difference across its two poles is as small as 0.1 at. %, the estimated velocity of DNP can reach 10~30 nm/s, which is in the same order as what was observed in experiments, ~50 nm/s.

Action taken

We have revised the Supplementary discussion III in Supplementary Materials accordingly.

REVIEWERS' COMMENTS

Reviewer #1 (Remarks to the Author):

I have no further comments.